

# Stochastic perturbations for parametrisation tendencies in a convection-permitting ensemble

Clemens Wastl[1], Yong Wang[1], Aitor Atencia[1], and Christoph Wittmann[1]

1Department of Forecasting Models, Zentralanstalt für Meteorologie und Geodynamik, Vienna, Austria

*Correspondence to*: Yong Wang (yong.wang@zamg.ac.at)

**Abstract.** A modification of the widely used SPPT (Stochastically Perturbed Parametrisation Tendencies) scheme is proposed and tested in a Convection-permitting – Limited Area Ensemble Forecasting system (C-LAEF) developed at ZAMG (Zentralanstalt für Meteorologie und Geodynamik). The tendencies from four physical parametrisation schemes are perturbed: radiation, shallow convection, turbulence and microphysics. Whereas in SPPT the total model tendencies are perturbed, in the present approach (pSPPT hereinafter) the partial tendencies of the physics parametrisation schemes are sequentially perturbed. Thus, in pSPPT an interaction between the uncertainties of the different physics parametrisation schemes is sustained and a more physically consistent relationship between the processes is kept. Two configurations of pSPPT are evaluated over two months (one of summer and another of winter). Both schemes increase the stability of the model and lead to statistically significant improvements in the probabilistic performance compared to the original SPPT. An evaluation of selected test cases shows that the positive effect of stochastic physics is much more pronounced on days with high convective activity. Small discrepancies in the humidity analysis can be dedicated to the use of a very simple supersaturation adjustment. This and other adjustments are discussed to provide some suggestions for future investigations.

## 1 Introduction

Stochastic physics schemes are used worldwide in many ensemble prediction systems (EPS) to represent uncertainties related to simplifications and approximations in the numerical model itself. Such uncertainties are defined as "model error" and arise from different sources such as computational constraints, incomplete knowledge of physical processes, uncertain parameters in parametrisations and from discretization methods. These errors range from large spatial scales (e.g. use of climatological aeorosol fields) to very small scales due to the use of parametrisations of unresolved processes such as the microphysics or turbulence scheme.

Stochastic parametrisation schemes produce an ensemble of perturbed members where each member sees a different, but equally likely stochastic forcing. They have been shown to significantly improve the reliability of weather forecasts (Sanchez et al., 2016; Leutbecher et al., 2017). Process-based stochastic approaches address sources of uncertainty in a particular parametrisation scheme (Plant and Craig, 2008; Bengtsson et al., 2013; Kober and Craig, 2016) while more general approaches treat uncertainty from a number of processes with one single scheme. The most popular method of the latter is the Stochastically Perturbed Parametrisation Tendencies scheme (SPPT) and has been developed at the ECMWF (European Centre





for Medium-Range Weather Forecasts; Buizza et al., 1999; Palmer et al., 2009). In SPPT a spectral pattern generator produces random noise with prescribed amplitude and correlations in time and space. This multiplicative noise is used to perturb model tendencies of temperature (T), water vapor (Q) content and wind (U, V). SPPT is operational at forecasting centres worldwide (e.g. ECMWF, U.K. Met Office, Japan Meteorological Agency, etc.). It has also been proven to work for some limited-area

models at the convection-permitting scale, such as the Météo-France Applications of Research to Operations at Mesoscale model (AROME-EPS; Bouttier et al., 2012) or the Weather Research and Forecasting model (WRF; Berner et al., 2015). SPPT improves the reliability of forecasts by reducing biases in the ensemble forecasts and yields a greater ensemble spread (Weisheimer et al., 2014; Leutbecher et al., 2017).

An often-mentioned shortcoming of the SPPT approach is the lack of physical consistency (Ollinaho et al., 2017; Leutbecher

et al., 2017). SPPT only perturbs the net physics tendencies inducing an inconsistency with fluxes computed from unperturbed tendencies (e.g. surface fluxes). This creates an energy imbalance where individual ensemble members no longer conserve energy. To avoid numerical instabilities based on this misbalance, a tapering function has been introduced to SPPT in the IFS (Integrated Forecasting System) model of ECMWF. It reduces the perturbations smoothly to zero in the boundary layer and in the stratosphere. However, this tapering function destroys the physical consistency because it assumes a reduced model error

in the lowest and topmost parts of the atmosphere.

Furthermore, the original SPPT generates only one single stochastic pattern which is applied to the parametrised net tendencies of model variables. This implies that the different schemes are perfectly correlated with each other and have the same error characteristics. This assumption is not always valid as demonstrated by Shutts and Pallares (2014). They have been shown, for example, that the uncertainty in the cloud and convection scheme is much higher than in the radiation scheme. Following this

discrepancy, Sanchez et al. (2016) have developed a method where a multiplicative noise with different standard deviations for different processes (e.g. gravity-wave drag, boundary layer scheme) is applied to the Unified Model (UM) of the Met Office. Decoupled perturbations among the different schemes increase the ensemble spread, especially in the tropics. However, a tapering function is still needed to ensure numerical stability.

Applying multiplicative noise to net physics tendencies, as in SPPT, implies that the uncertainty representation vanishes where

the total tendency is zero. This is also the case if the tendencies from different physics parametrisations are large but act in opposite directions. To overcome this problem, Christensen et al. (2017) have modified the SPPT scheme in the IFS model by perturbing the tendencies of the physics parametrisations with independent stochastic patterns. This perturbation is done at the end of each time step, so no interaction of the uncertainties between the schemes is considered. This limitation is addressed in the present paper.

In this study, we propose a modified SPPT approach in which the physical consistency between the different parametrisation schemes is kept. The details of two different versions of the developed scheme are described in Sect. 2. Section 3 contains a comparison of these schemes with the SPPT approach for two recent test periods (July 2016, January 2017). Standard probabilistic scores (spread, RMSE, CRPS, etc.) are used for surface and upper air variables. In Sect. 4 the effect of stochastic



physics is analyzed on days with strong convection over the Alpine test area and compared to days with stable conditions. Section 5 contains a summary of the results together with a discussion and the final conclusions.

## 2 Experimental design and methodology

### 2.1 The C-LAEF system

The ensemble forecasting system C-LAEF (Convection permitting Limited Area Ensemble Forecasting) has been developed at the Austrian national meteorological service ZAMG (Zentralanstalt für Meteorlogie und Geodynamik) and is based on the convection-permitting AROME model (Seity et al., 2011). AROME is under active development within the international NWP (Numerical Weather Prediction) consortium ALADIN (Aire Limitée Adaptation dynamique Développement InterNational, Termonia et al., 2018) and RC LACE (Regional Cooperation for Limited Area Modelling in Central Europe, Wang et al.,

2018). AROME has been operationally used at ZAMG since 2014. The model is run on a domain centered on Austria and covers the Alpine region (Fig. 1). It has a grid spacing of 2.5km, 90 vertical levels and a time step of 60 seconds. The non-hydrostatic dynamical kernel of AROME is identical to that developed for the ALADIN model (Bubnova et al., 1995; Benard et al., 2010). The AROME physics package is mainly adopted from the research model Meso-NH (Mascart and Bougeault, 2011) with the following main components: One moment bulk microphysical scheme ICE3 (using three prognostic ice and

hydrometeor classes; Pinty and Jabouille, 1998); statistical sedimentation of falling hydrometeor species after Bouteloup et al. (2011);  a 1D 1.5-order turbulence scheme (Cuxart et al., 2000); a mass-flux typed shallow convection scheme with turbulence closure (Pergaud et al., 2009); no deep convection scheme is needed because deep convection is assumed to be resolved by the dynamics; three-layer surface scheme SURFEX (Surface Externalisée, Masson et al., 2013) using a tile approach including sub-schemes for land, vegetation, town, sea and lake. The radiation scheme for AROME is taken from the ECMWF IFS model

where short-wave radiation is computed after Fouquart and Bonnel (1980) and long-wave using the Rapid Radiative Transfer Model (RRTM, Mlawer, 1997).

The C-LAEF ensemble comprises 16 members using the first 16 out of a total of 51 members of ECMWF-ENS (ensemble system of the ECMWF IFS model) for the boundary conditions. Coupling is done every three hours using a Davies relaxation scheme (Davies, 1976). Weidle et al. (2013) have shown that 16 members are a good compromise between ensemble spread

and computational costs. The ECMWF-ENS global ensemble system is operated on a cubic octahedral grid with about 0.2° horizontal resolution and 91 vertical levels. The members are created via a combination of ensemble data assimilation (Isaksen et al., 2010) and singular vectors (Leutbecher and Lang, 2013) for the initial state and by using SPPT and the Stochastic Kinetic Energy Backscatter (SKEB) method (Berner et al., 2009) during model integration.

Since the authors are only interested in the effect of stochastic physics, no extra initial or boundary condition perturbations are

applied on the C-LAEF side. For the same reason, no data assimilation is used in the experiments and surface uncertainty is not taken into account either. These assumptions are deemed acceptable because only the difference between stochastic physics



perturbation schemes are studied. The C-LAEF system is run once per day (00:00 UTC) with a forecast range of 30 hours and an output frequency of one hour.

## 2.2 Stochastic physics schemes

### 2.2.1 SPPT

The original SPPT stochastic physics scheme was initially developed by Buizza et al. (1999) for the IFS model of the ECMWF. Palmer et al. (2009) modified the scheme by introducing a spectral pattern generator. It creates a random 2D field with a prescribed standard deviation, temporal and spatial correlation length. In the IFS implementation, three independent random patterns with different correlation scales are used. They are designed to span the uncertainty at mesoscale, synoptic scale and planetary space and time scales. The resulting random patterns are Gaussian distributed with zero mean, unit variance and a

homogeneous and isotropic horizontal autocorrelation. The amplitude of the perturbations is restricted to a range defined by the standard deviation $[-2\sigma, 2\sigma]$. The net tendencies, P, of wind (U and V component), temperature (T) and water vapor content (Q) are multiplied at each time step during the model integration with this perturbation field to generate the perturbed physics tendencies. The perturbed net tendency of the physics parametrisations (P') is represented by:

$$P' = (1 + \alpha r) \sum_{i=1}^{n} P_i \qquad (1)$$

where $\alpha$ is a level dependent constant defined by the tapering function, r is a random number defined by the perturbation pattern (in the IFS composed from a linear combination of three independent patterns with different correlation scales), $P_i$ is the unperturbed tendency of one parametrisation scheme and n is the number of physics schemes contributing to the total tendency equation. The first row in Fig. 2 illustrates the way how the physics tendencies of C-LAEF are perturbed in SPPT. Due to the multiplicative feature, the scheme attributes the greatest uncertainties to the areas where the largest net tendencies

P occur. The shape of the tapering function $\alpha$ can be controlled in the model setup. It reduces the perturbations to zero in the boundary layer below 900 hPa (default) and in the stratosphere above 100 hPa (default). $\alpha$ is set to 1 for all remaining levels, thereby retaining the vertical structure that results from the physics parametrisations. The tapering function has been introduced to the IFS model to avoid numerical instabilities - it is not necessary in some regional models (e.g. WRF, COSMO). Bouttier et al. (2012) have successfully implemented SPPT in the AROME model. Some changes have to be made to the

original SPPT in order to adapt the methodology from IFS to AROME. The main change is the adaption of the spectral pattern generator from the spherical harmonics applied in the IFS to the biFourier functions used in AROME. The link between the variance spectrum and the biFourier representation follows the formulation by Berre (2000). At the edges of the model domain, the uncertainties originate only from the lateral boundary formulation and the physical tendencies are smoothly relaxed to zero. Due to the relatively short forecast range of the convection permitting AROME model (30 hours) only one stochastic pattern

is used, instead of three in case of the IFS model.



In the AROME implementation of SPPT, no perturbations of temperature and humidity are applied if the resulting humidity value is negative or exceeds the critical saturation value (supersaturation adjustment, Bouttier et al., 2012). This is different from the IFS version, where a smooth humidity reduction is applied in such cases (Palmer et al., 2009). The default settings of the pattern generator applied by Bouttier et al. (2012) have to be tuned to the C-LAEF configuration.

This is done using a two-week test period characterized by high convective activity (16 − 30 July 2011). The goal of this optimization is to generate a realistic spread without creating a model bias. Using SPPT in the AROME model requires a tapering function to avoid numerical instabilities. Experiments with tapering off in the boundary layer in SPPT resulted in several model crashes during the test period because of too strong wind over the Alps. However, this has not been further investigated.

The main characteristic of this scheme, described as "SPPT" hereinafter, is the perturbation of net tendencies without considering the contribution of each individual physics tendencies (Fig. 2). In other words, this approach assumes that no uncertainty is added when the net tendency is zero, even though the single physics schemes might have large but compensating contributions.

### 2.2.2 Physical parametrisation based SPPT (pSPPT)

The restrictions and assumptions made in the original SPPT approach, which have produced unsatisfactory results within the Austrian domain, have led to the idea of setting up a modified version of SPPT. The main goal is to maintain the interactions between the individual physics schemes, and thus, to keep the model stable. The different physics schemes in AROME are called subsequently in the following order: radiation, shallow convection, turbulence and microphysics. Each scheme provides a partial tendency of the main model quantities T, U, V, Q. The condensed water species are not diretlcy perturbed, they are

adjusted at each time step by the fast microphysics step (Seity et al., 2011). In the original SPPT version the partial tendencies of the different physics parametrisations are summed up at the end of the time step and this net tendency is finally perturbed by the noise of the pattern generator as in Eq. (1). As a consequence, the uncertainties resulting from one scheme are not passed to the following scheme.

In the present study, it is proposed to perturb the partial tendencies of the physics schemes separately and to consider the

resulting perturbed fields in the subsequent physics scheme. We call this approach physical parametrisation based SPPT (pSPPT hereinafter). Equation (2) shows the formulation of the perturbed partial tendency of each parametrisation scheme in this new pSPPT scheme, an illustration of this is given in Fig. 2. Each random pattern ($r_i$) is generated separately by the pattern generator using a different seed.

$$P_i' = (1 + \alpha r_i) P_i \quad for \ i = 1, n \qquad (2)$$

The uncertainties are passed through the different schemes and as a consequence, the issue of only perturbing non-zero net tendencies is avoided. For example, if the turbulence scheme provides a strong positive temperature tendency and the microphysics scheme a comparable negative temperature tendency, no effect of stochastic physics perturbations is present in



the original SPPT. However, pSPPT will either intensify or weaken the strong positive tendency of the turbulence scheme, depending on the stochastic pattern. The resulting tendency is then processed in the microphysics scheme and afterwards again adapted by the perturbation process. This approach has a positive effect on the stability of the model, as shown by a reduction of the number of model crashes in a sensitivity study during the 2011 test period. The increased numeric stability in pSPPT

allows the tapering function for microphysics, radiation, and shallow convection schemes to be switched off, being only maintained for the turbulence scheme. In the turbulence scheme, the stochastic perturbations in the lower atmosphere produce too much instability and therefore the model crashes after some time steps.

A potential drawback of the pSPPT approach is a possible duplication in attributing errors across schemes which can introduce inherent correlations between the perturbations applied to one physics scheme and the output of a later scheme (Christensen

et al., 2017).

### 2.2.3 Independent physical parametrisation based SPPT (ipSPPT)

In pSPPT as well as in SPPT, the tendencies of all considered variables (T, U, V and Q) are perturbed with the same stochastic pattern, which assumes that the different variables in the parametrisation schemes have similar error characteristics. However, this assumption is vague and might not always be satisfied as Boisserie et al. (2013) have shown. This leads us to a new

approach where the tendencies resulting from the physical parametrisation schemes (temperature, wind components and water vapor content) are perturbed by individual stochastic patterns. It can be seen as an adaptation of the pSPPT approach presented before and is called ipSPPT hereinafter. Equation (3) highlights the independence of this ipSPPT methodology, by formulating the perturbation of T, U, V and Q separately. An illustration of this is given in the last row of Fig. 2.

$$T_i' = \left(1 + \alpha r_{i,1}\right) T_i; \quad U_i' = \left(1 + \alpha r_{i,2}\right) U_i; \quad V_i' = \cdots \quad for\ i = 1, n \qquad (3)$$

As a consequence, the random field applied to e.g. the temperature tendency (T) is different from the one used for the wind components (U, V) or the water vapor content (Q). Tapering is treated in ipSPPT as in the pSPPT approach (active only for the turbulence scheme).

The first SPPT version in the IFS model (Buizza et al., 1999) has also used such separate patterns for the different parametrised tendencies. However, it has been removed in the revised SPPT scheme (Palmer et al., 2009) because some physical

relationships within a parametrisation scheme could be violated in this way (see Sec. 5).

### 2.3 Experimental set-up and verification methods

A two-week period (16 – 30 July 2011) is used to optimize the settings of the spectral pattern generator and the different parameters of the stochastic physics schemes in the C-LAEF system. A set of four experiments has been chosen for a long-period verification: One experiment without any stochastic physics perturbations (REF), one containing the original SPPT

approach (SPPT – Sect. 2.2.1), a version using physical parametrisation based SPPT (pSPPT – Sect. 2.2.2) and a version of



pSPPT with independent patterns for the prognostic variables (ipSPPT – Sect. 2.2.3). The experimentation is conducted over a summer month (July 2016) and winter month (January 2017) with one run per day (00:00 UTC) and 30 hours forecast range. The model domain is shown in Fig. 1 and corresponds to the operational deterministic AROME domain used at ZAMG.

The upper-air weather variables are verified using ECMWF analyses at the 500 hPa and 850 hPa levels, while surface

variablese are verified using SYNOP station data. Forecast values are interpolated to the observation location for smooth fields such as 2 m temperature, 10 m wind speed or surface pressure. In the case of precipitation, the forecasts are matched to the nearest grid point. A height correction is applied to the 2 m temperature to account for discrepancies between model surface and station height. The verification is performed over the whole C-LAEF domain in Fig. 1 which contains more than 1200 observation sites. Beside classical scores such as ensemble spread, ensemble bias or ensemble root-mean-square error (RMSE),

the skill of the forecasts is also evaluated by a set of probabilistic scores like Continuous Ranked Probability Score (CRPS; Wilks, 2011) or the Brier Score (BS; Hamill and Colucci, 1997). The statistical significance of the score differences between the three experiments and the reference run is defined by using a bootstrapping confidence test. Therefore a block of three days is sampled out of the 31-day verification period (summer and winter, respectively) and the time averaged score difference to the reference run is computed. An empirical distribution of all three experiments is constructed by repeating this procedure

for 5000 times. The score difference is deemed significant if its sign is not contradicted by more than 10% of the sample (for more details see Wilks, 2011).

## 3 Results

### 3.1 Summer period: July 2016

#### 3.1.1 Upper air verification

The large-scale synoptic pattern in the first half of July 2016 was characterized by a very deep trough over the British Islands directing an extensive southwesterly flow over the target area of Central Europe. This arrangement resulted in a strong advection of warm and moist air masses towards the Alps leading to strong convective activity. Numerous thunderstorms causing local flash floods and even tornadoes were observed during this time. In the second part of July 2016 a very weak pressure gradient was established over Central Europe causing some isolated convection with stationary thunderstorms and

locally high precipitation amounts.

The first two rows in Fig. 3 show the performance of the three experiments (SPPT, pSPPT, ipSPPT) as a difference relative to the reference run without any stochastic physics, for temperature (first row) and wind speed (second row) at 500 hPa (first column) and 850 hPa (second column), respectively. The use of stochastic physics should result in an increase of ensemble spread together with an unmodified, or sometimes reduced model error (Leutbecher et al. 2017). Hence, positive differences

in spread and negative differences in RMSE are desirable.

Significant differences are represented by filled circles for ensemble spread and by crosses for RMSE in Fig. 3. The ipSPPT experiment (black) shows the highest gain in spread for both temperature and wind speed at both levels. The original SPPT

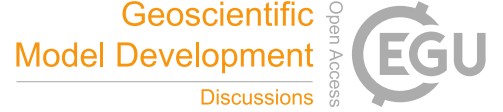



(red) and the pSPPT approach (blue) also exhibit an increase of spread. Focusing on the RMSE (dashed lines), Fig. 3 reveals a small increase of RMSE for temperature at 500 hPa in all three experiments, especially from forecast hour twelve onwards. For both pSPPT and ipSPPT, this temperature increase is even statistically significant. Interestingly, this feature is not present at 850 hPa, where the use of stochastic physics leads to a general decrease of RMSE. A slight temperature increase above 800

hPa has already been observed by Bouttier et al. (2012) in the French AROME-EPS experiment, but no explanation was provided. This effect can partly be explained by the very simple supersaturation adjustment which is used in our experimentation, but this needs to be further investigated over a longer test period. Perturbations are not applied to temperature and water vapor content when the saturation level is exceeded. Hence, a general trend towards a systematic drying of the atmosphere is implied, because more negative perturbations are applied in total. This drying effect has already been highlighted

by several SPPT studies (Berner et al., 2009; Bouttier et al., 2012). To overcome this shortcoming, Davini et al. (2017) have developed a moisture conservation fix which has also been adopted to the global IFS model by Leutbecher et al. (2017). An improved supersaturation adjustment has also been developed for the AROME model by Szucs (2016), but it has not yet been implemented to the present experimentation. Szucs (2016) evaluated this drying effect for the AROME-EPS model during the convective season in 2015. After 24 h lead time the use of a simple supersaturation adjustment resulted in a negative bias for

relative humidity of about 1% at 700 hPa and about 2% at 850 hPa and at the surface. In terms of temperature, the simple supersaturation is translated in a slightly positive temperature increase due to the omission of negative temperature perturbations when the supersaturation level is reached. This temperature effect is not present at lower levels, because the reduced humidity at the surface is compensated by stronger evaporation during the day and rapidly decreasing temperatures during the night (Leutbecher et al., 2017).

The behavior of the C-LAEF system is indicated by the third row in Fig. 3 where the absolute spread and RMSE for temperature and wind speed at 850 hPa is shown. The RMSE is generally high, even at initialization time, because these simulations are pure downscaling of the IFS model without any data assimilation. The spread increases with lead time, while the RMSE is higher during the day when radiation and turbulent fluxes are larger and convection occurs. A spread smaller than the RMSE is an indicator of an under-dispersive ensemble. The spread and RMSE lines are closer in the ipSPPT experiment, showing the

positive effect of this method on the ensemble performance.

This behavior is also reflected in the probabilistic CRPS (not shown). CRPS measures the skill of the ensemble mean forecast as well as the ability of the perturbations to capture the deviations around it (Bowler et al., 2008). A low value of CRPS indicates a more skillful forecast. For temperature at 850 hPa and wind speed at both 850 hPa and 500 hPa, the application of the stochastic physics methods leads to a significant decrease of CRPS, compared to the reference run. Only for temperature

at 500 hPa the CRPS difference is slightly positive for all three experiments due to the positive temperature bias. CRPS shows a diurnal cycle similar to RMSE in Fig. 3.



### 3.1.2 Surface verification

The same verification is done for the surface variables 2 m temperature, 10 m wind speed, mean sea level pressure (MSLP) and precipitation. Spread and RMSE plots are not shown, but CRPS is shown in the first four panels of Fig. 4. For temperature and wind speed all three stochastic physics experiments have smaller CRPS values representing a more skillful forecast. This

behavior can be explained by an increase of the ensemble spread while the model error is not noticeable influenced by the stochastic physics perturbations (not shown). The increase of spread is smallest for the SPPT experiment which can be attributed to the tapering function in the boundary layer, which is used for all parametrisation schemes in this experiment. Mean sea level pressure (MSLP) in the original SPPT and pSPPT does not show a noticeable impact, but in ipSPPT there is a significant improvement. The ipSPPT results in an improvement in the precipitation verification (reduced CRPS) as well,

which is especially significant in the afternoon, when convection is abundant during summer season (Fig. 4). The significant reduction of CRPS for precipitation is mainly caused by a large increase of ensemble spread (not shown).

To investigate the effect of the simple supersaturation treatment in the boundary layer, 2 m temperature and relative humidity biases relative to the REF experiment are given in Fig. 4 (last row). It reveals a general trend towards lower temperatures in all experiments with stochastic physics and the strongest effect for the ipSPPT experiment in the afternoon and evening hours.

A significant drying of the boundary layer is obvious in all three experiments with stochastic physics and can be attributed to the simple supersaturation adjustment.

Generally, the differences in the scores analysed in this section are quite small but significance is reached and they are comparable to other studies of stochastic physics on the convection-permitting scale (e.g. Bouttier et al., 2012; Bowler et al., 2008).

## 3.2 Winter period: January 2017

### 3.2.1 Upper air verification

January 2017 was the coldest January in the last 30 years in most parts of Austria. The weather situation during the first two weeks was characterized by a widespread high-pressure system over the eastern Atlantic Ocean blocking the westerlies and enabling the advection of cold polar air masses from the Arctic Sea towards Central Europe. Embedded fronts caused strong

snow falls resulting in an area-wide snow cover over Central Europe. This situation fueled the local production of cold air near the surface during the long winter nights. In the second part of the month, a high-pressure system over Scandinavia caused easterly winds over the Alps advecting extremely cold, continental air masses from Russia into the target domain.

Compared to the summer period verification, the scores of upper air variables of January 2017 in Fig. 5 are much smaller. For temperature and wind speed at both levels (500 hPa and 850 hPa) the use of stochastic physics results in an increase of ensemble

spread. However, statistical significance over the whole forecasting range is only reached for temperature and wind speed at 850 hPa in the ipSPPT approach. RMSE is not influenced significantly, except for the wind speed at 850 hPa in case of ipSPPT. However, a small trend towards higher temperatures and lower humidity in the experiments with stochastic physics also



persists in winter (not shown). CRPS at upper air is slightly decreased for all variables considered in January 2017, but being statistically significant only in case of ipSPPT (not shown). It seems that the different error representations of the model variables T, U, V and Q have a positive effect on the scores at these levels in winter.

### 3.2.2 Surface verification

The RMSE of the surface variables in C-LAEF is very large for January 2017 (last row of Fig. 6). The bias is strongly positive, especially for 2 m temperature, indicating significantly higher temperatures in the model than observed. This can be partly explained by the fact that data assimilation is not used. However, other operational models at ZAMG also performed poorly during this period, with the pronounced temperature inversions in Alpine valleys posing big problems for the models. C-LAEF simulated a breakup of the temperature inversion in the afternoon, but in reality, the cold air was very persistent.

The ensemble spread is much smaller than the model error showing a highly underdispersive ensemble. This fact can be explained by the absence of initial conditions and surface perturbations in our experimentation. Focusing on the improvements compared to the reference ensemble, the first two rows of Fig. 6 show an increase of ensemble spread for the ipSPPT and especially pSPPT experiment, while the original SPPT method does not have a strong effect. This can be attributed to the stronger tapering in SPPT. pSPPT also produces a significant increase of the RMSE for temperature around noon (+ 12 h).

Finally, the effect of the simple supersaturation adjustment, which influences the scores in the summer period, is not visible at the surface in January 2017. This is because January 2017 was a rather dry month, with a lot of sunny days where saturation was rarely reached in the lower atmosphere.

The 10 m wind speed exhibits an increase in spread for all three experiments, while the model error is barely modified. In the ipSPPT experiment, the RMSE of the mean sea level pressure is significantly decreased which is also reflected in a negative

CRPS difference (not shown). The other two experiments instead reveal a RMSE increase compared to REF. For precipitation, the ensemble spread is significantly increased in the ipSPPT experiment and partially in the pSPPT scheme. The RMSE of precipitation is decreased for all three experiments between 12 and 24 hours lead time, compared to REF.

### 4 Impact on convection

Forecasting convection in summer still remains one of the biggest challenges for the current high-resolution NWP systems,

especially in complex terrain like the Alps. Section 3 showed that pSPPT and especially ipSPPT can significantly improve the ensemble spread of precipitation forecasts in summer. To further investigate this behavior, several test cases with high convective activity are selected out of the July 2016 period and compared to days with stable conditions. The selection of cases is based on the Convective Available Potential Energy (CAPE) and the observed precipiatation gained from the operational analysis system INCA (Haiden et al., 2011; Wang et al., 2017). All days with CAPE > 1000 Jkg$^{-1}$ in the afternoon (15:00 UTC)

averaged over the whole INCA domain (Fig. 1) and some observed thunderstorms are grouped into the convective class, days with CAPE < 500 Jkg$^{-1}$ remain in the non-convective class. Following this classification, 13 days of July 2016 can be assigned to the convective class and ten days to the non-convective class.





Figure 7 shows the ensemble spread and RMSE for precipitation of all experiments relative to an ensemble without stochastic physics (REF). For this precipitation verification the observations are taken from the INCA analysis system which combines rain gauge and radar data on a 1 km grid. Comparing the two columns of Fig. 7 reveals a much stronger impact of stochastic physics on the ensemble spread at days with significant convection. Especially for the ipSPPT approach the spread increase

(compared to REF) in the afternoon of convective days is about five times higher than for days with stable conditions. Also for SPPT and pSPPT the spread increase is mainly restricted to days with convection. The effect on RMSE of precipitation is generally smaller (see also Sect. 3.1.2). A slight reduction of RMSE in the afternoon can be seen for SPPT and pSPPT with the larger values at convective days. The effect on RMSE for the ipSPPT experiment is generally small in both cases. This case study shows that introducing perturbations into a model is much more effective when convection and vertical motion

in the atmosphere is high. This is only shown for precipitation in Fig. 7, but also for temperature or wind speed the effect of stochastic physics is much higher at convective days (not shown).This explains why the scores presented in Sect. 3 are generally smaller in winter when the conditions in the considered area are generally much more stable than in summer.

## 5 Discussion and conclusions

In this study we have proposed two physical parametrisation based SPPT versions (pSPPT, ipSPPT) and have investigated

their performance in a convection-permitting ensemble for one summer and one winter month. In pSPPT the partial tendencies of turbulence, radiation, shallow convection and microphysics are perturbed individually and interact with the subsequent parametrisation schemes. In other words, each parametrisation sees the updated state including the perturbed tendencies of the previous parametrisations (Fig. 2). In ipSPPT an independent perturbation is additionally applied to the parametrisation tendencies T, U, V and Q. These two schemes have been compared to the original SPPT method (Buizza et al., 1999; Bouttier

et al., 2012) and the control ensemble without any stochastic perturbations. As expected, the use of stochastic physics increases the ensemble spread, especially in periods with high convective activity (summer period). The gain of spread is clear in temperature and wind speed at all model levels, with the highest increase near the surface. This can be mainly attributed to the reduced tapering of perturbations in the boundary layer in pSPPT and ipSPPT. In the case of precipitation, SPPT has little effect on the ensemble spread, whereas the new ipSPPT scheme reveals a significant increase. The model error has been

analyzed by calculating the RMSE of each experiment as difference to the reference run. For most variables stochastic physics lead to a slight decrease of model error throughout all lead times. The strongest effect is observed with the ipSPPT approach. In the case of temperature, the effect is much more complex: a temperature positive related bias is oberved in the upper levels (e.g. 500 hPa) while a negative difference of bias is obtained near the surface. The simple supersaturation adjustment used in our experimentation has a strong impact on the temperature and especially humidity scores presented here. This adjustment

tends to favor positive temperature and negative water content perturbations due to omitting perturbations when supersaturation is reached. This leads to a significant drying of the atmosphere which results in a cooling effect in the surface boundary layer due to higher evaporation rates during the day and stronger long-wave emission at night. These problems



should be reduced by using an improved supersaturation adjustment which has already been developed for the AROME model (Szucs, 2016). However, this has not yet been used in the present study, but will be tested in the near future.

CRPS confirmed the better performance of the ensemble when using stochastic physics perturbations. These improvements are generally much smaller in winter than in summer, which can be explained by the more stable stratification of the atmosphere. A small temperature increase is sufficient to trigger convection and to influence wind, humidity and precipitation fields in summer. This conclusion is supported by a more in depth analysis of a convective events presented in this paper.

The main reason for trying two new approaches of stochastic physics perturbations are the restrictions and assumptions made in the original SPPT. The first assumption is the use of a tapering function which has been implemented to SPPT to consider the imbalance between perturbed atmospheric tendencies and the unperturbed surface fluxes and thus to avoid numerical instabilities. On the other hand, smoothly relaxing the perturbations to 0 in the lowermost levels of the atmosphere implies a different error representation in the vertical which can be considered critical. Sensitivity studies during the test period of July 2011 with switching off tapering in the SPPT approach, showed about 10% of model crashes due to exceptional high wind speeds over the Alps. Perturbing the physical schemes separately and considering this perturbed fields in the subsequent parametrisation (pSPPT) results in a positive effect on the stability of the model. In this case the tapering function could be switched off for microphysics, radiation, and shallow convection without any problems. For the turbulence scheme, the perturbations in the lower atmosphere produce too much instability especially in the Alps, and therefore the tapering function has to be turned on. Switching off tapering function separately for the schemes is only possible in the new, independent approaches with partial tendencies (pSPPT, ipSPPT). In the case of the original SPPT, the physical schemes cannot be influenced independently.

The main difference between the pSPPT approach presented here and the independent SPPT (iSPPT) method proposed by Christensen et al. (2017) is the time when the perturbations are applied. In iSPPT the stochastic perturbations are applied at the end of the time step, whereas in the approaches presented in this paper, perturbations are applied directly after each parametrisation. Hence, an interaction of of the uncertainty of one physical scheme in the subsequent one is considered in pSPPT and ipSPPT which seems to increase the consistency of the model, but this needs to be confirmed using longer experiments. Of course, sequentially perturbing the partial tendencies implies a possible duplication of model error representation (Christensen et al., 2017). However, the results in Sect. 3 have shown that a significant increase of spread goes along with only a small effect on the model error (RMSE) when applying pSPPT (ipSPPT). A direct comparison of the pSPPT and iSPPT approach within the C-LAEF framework would be very interesting at this point, but it is beyond the scope of this paper and is planned in a future study.

The very flexible structure of the pSPPT approach also allows a combination with other uncertainty representations such as the parameter perturbations scheme in Ollinaho et al. (2017).

The ipSPPT approach is a modification of pSPPT where the tendencies of the variables T, U, V, and Q receive separate perturbations. As shown in Sect. 3, this approach obtains the best probabilistic scores overall, even though the method is considered critical from a physical point of view. A major concern with the ipSPPT approach is that the balance between the



quantities resulting from one parametrisation scheme can be disturbed (Palmer et al., 2009). For example, the microphysics scheme provides an increase of temperature at a certain point due to condensation processes which are also decreasing the water vapor content. This equilibrium is destroyed if temperature and water vapor content tendencies are perturbed with opposite signs. On the other hand, it cannot be assumed that T and Q have exactly the same error characteristics, as it is

supposed in SPPT and pSPPT. Furthermore, in SPPT and pSPPT the wind direction is never altered stochastically, since the tendencies of the U and V components are always using the same stochastic pattern. Testing over a longer period will be necessary to identify if conservation rules are violated in ipSPPT and if it is really applicable in an operational framework. Last but not least, perturbations in SPPT are only active in areas where the net tendency is not 0, even though the individual physical parametrisation schemes might have strong opposite contributions. This shortcoming is avoided by perturbing the

partial tendencies of the physics parametrisations in both pSPPT and ipSPPT.

In our experiments no ensemble data assimilation or errors in the initial conditions are taken into account. Consequently, only the impact of different stochastic physics approaches compared to a reference ensemble has been considered. The focus on relative scores between the different experiments justifies somehow also the fact that we did not consider observation error simulations in our verification. Of course, including observation error can have a strong impact on scores like ensemble spread

(Bouttier et al., 2012), but we suppose that it would act in the same direction for all experiments and therefore the relative conclusions stay the same.

The next step in the development of C-LAEF is to introduce the new stochastic perturbation schemes to a full system with data assimilation and initial perturbations. The verification in this operational framework will show the operational benefit of these new approaches for the C-LAEF system.

**Code and/or data availability**

The C-LAEF and AROME codes including all related intellectual property rights, are owned by the members of the LACE consortium and ALADIN consortium. Access to the ALADIN-LAEF and AROME systems, or elements thereof, can be granted upon request and for research purposes only. INCA data are only available subject to a licence agreement with ZAMG.

**Author contribution**

Clemens Wastl developed the different stochastic schemes together with Yong Wang. Christoph Wittmann designed the experiments and carried them out together with Clemens Wastl. Aitor Atencia was responsible for the verification of the results. Clemens Wastl prepared the manuscript with contributions from all co-authors.

**Competing interests**

The authors declare that they have no conflict of interest.



## Acknowledgments

The authors gratefully thank all of the colleagues who contributed to this study. Special thanks go to Eric Bazile and Yann Seity from Météo-France for their input into this work through discussions and to the ECMWF for the possibility to run all the experiments on their supercomputer.

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

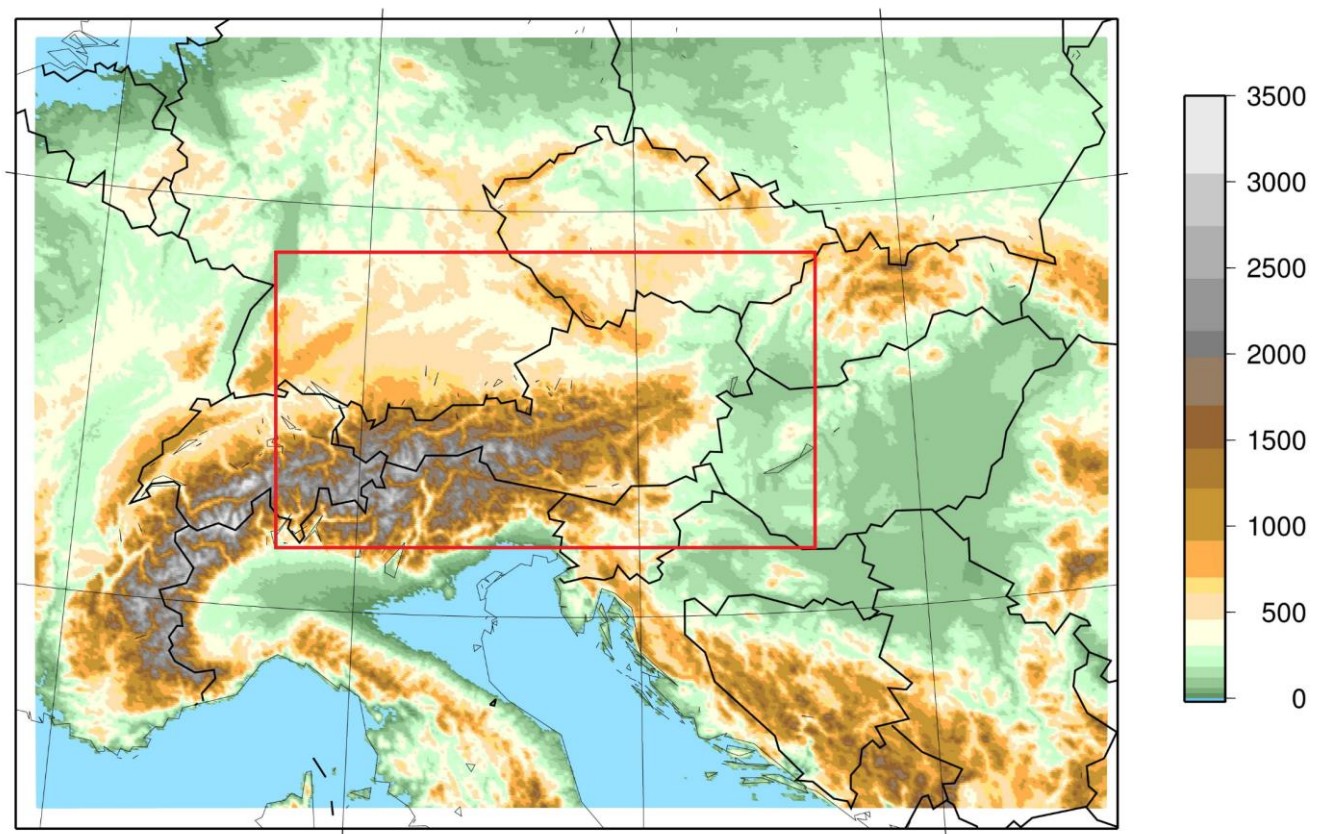

**Figure 1. Domain of the C-LAEF system including the INCA domain for precipitation verification (red). The colouring shows the altitude (m).**



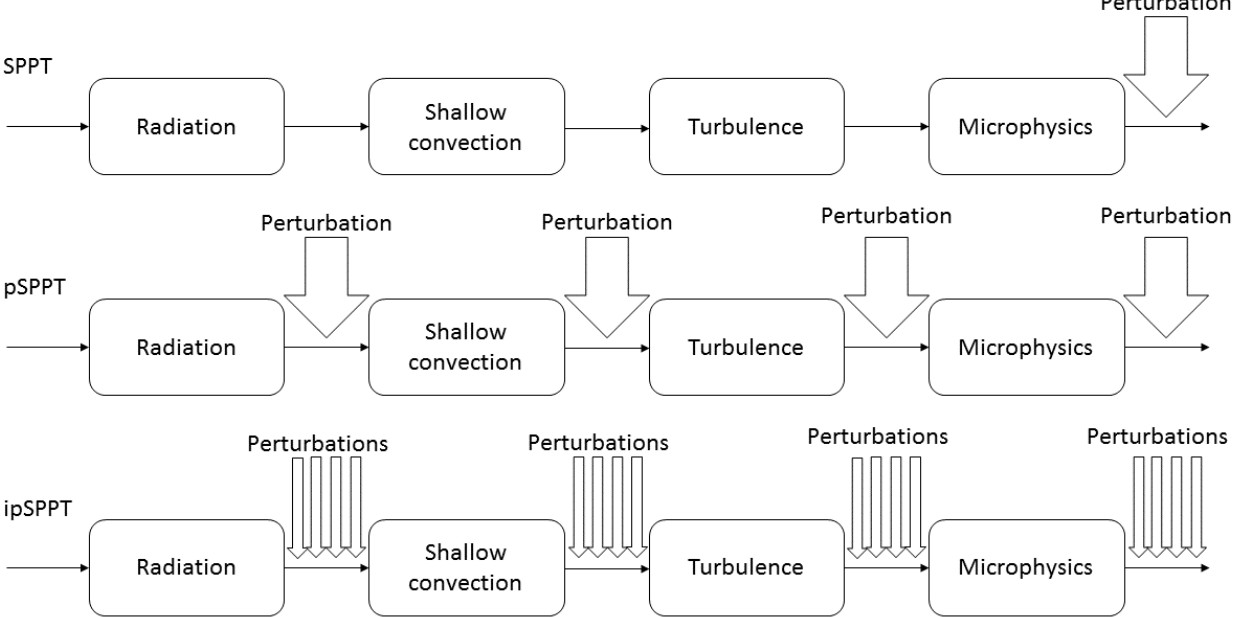

Figure 2. Illustration of how the stochastic perturbations are applied in the different physics parametrization schemes of SPPT (first row), pSPPT (second row) and ipSPPT (last row).

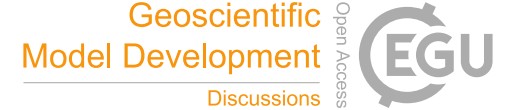

**Figure 3.** Ensemble spread (solid lines) and RMSE (dashed lines) as a function of lead time for temperature (K) and wind speed (m s-1) at 500 hPa and 850 hPa in July 2016. Scores in the first two rows are shown as difference to an ensemble without any stochastic physics (REF), circles (crosses) denote significant differences for the ensemble spread (RMSE). The last row shows absolute numbers for all four experiments at 850 hPa.





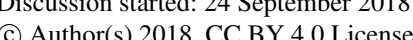

**Figure 4. The upper two rows show the Continuous Ranked Probability Score (CRPS) as a function of lead time for surface variables 2 m temperature (K), 10 m wind speed (m s-1), mean sea level pressure (MSLP, hPa) and precipitation (mm) in July 2016. The lowest row shows the BIAS of 2 m temperature (K) and relative humidity (%) for the same period. All numbers are shown as a difference to C-LAEF without any stochastic physics (REF). Circles denote significant differences of CRPS and BIAS, respectively.**



**Figure 5. Ensemble spread (solid lines) and RMSE (dashed lines) as a function of lead time for temperature (K) and wind speed (m s-1) at 500 hPa and 850 hPa in January 2017. Scores are shown as difference to an ensemble without any stochastic physics (REF), circles (crosses) denote significant differences for the ensemble spread (RMSE).**





**Figure 6. As in Figure 3, but for surface variables 2 m temperature (K), 10 m wind speed (m s-1), mean sea level pressure (MSLP, hPa) and precipitation (mm) in January 2017. The last row shows absolute numbers for temperature and precipitation.**



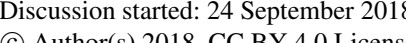

**Figure 7. Ensemble spread (upper row) and RMSE (lower row) of precipitation (mm) as a function of lead time. The left column refers to days with high convective activity, the right column to days with stable conditions. Scores are shown as difference to an ensemble without any stochastic physics (REF).**

