# Peer review of "Independent perturbations for physics parametrisation tendencies in a convection permitting ensemble (pSPPT)"

_Geoscientific Model Development, 2018_

## Short Comment (SC1) · 22 Oct 2018

Dear authors,

in my role as Executive editor of GMD, I would like to bring to your attention our Editorial version 1.1:

http://www.geosci-model-dev.net/8/3487/2015/gmd-8-3487-2015.html

This highlights some requirements of papers published in GMD, which is also available on the GMD website in the 'Manuscript Types' section:

http://www.geoscientific-model-development.net/submission/manuscript_types.html

[Figure]

In particular, please note that for your paper, the following requirement has not been met in the Discussions paper:

- "The main paper must give the model name and version number (or other unique identifier) in the title."

Please add at least the acronym of your new parametrisation (pSPPT) in the title of your manuscript upon revision of the article. Additional, if applicable, add a version number (the method might be changed in the future and those methods should be distinguishable).

In addition, regarding the code availability section, in case it is possible for the user to obtain a license, please add further information on how to obtain it. Be aware, that in all cases the exact version of your method published here, must be kept.

Yours,

Astrid Kerkweg

---

## Referee Comment (RC1) · M. Denhard (Referee) · 26 Oct 2018

General comments 1. Does the paper address relevant scientific modelling questions within the scope of GMD? Yes. It concerns parametrizations in convection permitting NWP systems. 2. Does the paper present novel concepts, ideas, tools, or data? Yes. Especially the ipSPPT scheme is a quite innovative idea. 3. Does the paper represent a sufficiently substantial advance in modelling science? In general, the generation of spread with stochastic physics schemes in NWP ensembles is quite hard work, but the paper presents some nice improvements. 4. Are the methods and assumptions valid and clearly outlined? Yes. 5. Are the results sufficient to support the interpreta-

tions and conclusions? Yes. 6. Is the description sufficiently complete and precise? Yes. 7. Do the authors give proper credit to related work and clearly indicate their own new/original contribution? Yes. 8. Does the title clearly reflect the contents of the paper? In its present form the title does not really reflect that stochastic perturbations are independently applied to parametrization schemes and prognostic variables. 9. Does the abstract provide a concise and complete summary? Yes. 10. Is the overall presentation well structured and clear? Yes. 11. Is the language fluent and precise? Yes. 12. Are mathematical formulae, symbols, abbreviations, and units correctly defined and used? Yes. 13. Should any parts of the paper (text, formulae, figures, tables) be clarified, reduced, combined, or eliminated? Figures 3-7 must be substantially improved. There are no ordinate labels and no references to the different panels in the figure caption. 14. Are the number and quality of references appropriate? Yes. 15. Is the amount and quality of supplementary material appropriate? Yes.

Specific comments

— Abstract: "Both schemes . . . lead to statistically significant improvements in the probabilistic performance compared to the original SPPT". The authors have shown statistical significance of the different schemes with respect to the reference experiment of no perturbations not with respect to SPPT (see also Page 7, lines 11-12: "The statistical significance of the score differences between the three experiments and the reference run is defined by using a bootstrapping confidence test.")

— Page 4, line 22: "The tapering function . . . - it is not necessary in some regional models (e.g. WRF, COSMO)." Are there any references? Do you know why?

— Page 6, lines 8-10: "A potential drawback of the pSPPT approach is a possible duplication in attributing errors across schemes which can introduce inherent correlations between the perturbations applied to one physics scheme and the output of a later scheme (Christensen et al., 2017)." In contrast to SPPT the pSPPT approach enables switching of the tapering function. Christensen et al. 2017 state about the

effects of tapering in SPPT "... this method cannot represent uncertainty in the vertical distribution of convective heating. SPPT does not perturb fluxes at the surface or top of atmosphere, introducing inconsistencies between the perturbed tendencies in a column and these fluxes." When balancing the disadvantages from tapering inconsistencies or a possible duplication in attributing errors across schemes, it seems not unreasonable to try out the pSPPT approach. Moreover, it could be expected that errors naturally appear all along the production line of the parametrization schemes and will be processed through the different schemes anyway, presumably being inherently correlated at the output. The main fundamental reason for keeping the parametrization chain deterministic and balanced is that in deterministic parametrisations similar tendencies on input produce similar outputs and thus large scale correlations of the input tendencies on the grid scale will not be altered by the parametrization chain. Therefore, in SPPT only the final output of the chain is perturbed with large scale stochastic patterns which are tied to resolved physical processes on the grid scale. Christensen et al., 2017 state: "SPPT also imposes large spatio-temporal correlation scales when perturbing tendencies to represent the correlation of model uncertainties in space and time, but these correlation scales have not been measured and are not tied to physical processes." Because the correlation length scales of the stochastic patterns are the same one can still expect that the correlations of the input tendencies survive not only the SPPT but also the intermediately perturbed parametrization chain of pSPPT.

— Page 12, lines 13-15: "Perturbing the physical schemes separately and considering this perturbed fields in the subsequent parametrisation (pSPPT) results in a positive effect on the stability of the model. In this case the tapering function could be switched off for microphysics, radiation, and shallow convection without any problems." Page 12, lines 23-24: "Hence, an interaction of of the uncertainty of one physical scheme in the subsequent one is considered in pSPPT and ipSPPT which seems to increase the consistency of the model," What does consistency of the model means? Does it mean that it runs more stable? It is a little bit surprising that the tapering function could be switched of in the ipSPPT as well, because there is much more uncorrelated noise

than in the two other schemes? Where does the stability comes from?

— Pages 12-13, lines 32-6: "The ipSPPT approach is a modification of pSPPT where the tendencies of the variables T, U, V, and Q receive separate perturbations. As shown in Sect. 3, this approach obtains the best probabilistic scores overall, even though the method is considered critical from a physical point of view. A major concern with the ipSPPT approach is that the balance between the quantities resulting from one parametrisation scheme can be disturbed (Palmer et al., 2009). For example, the microphysics scheme provides an increase of temperature at a certain point due to condensation processes which are also decreasing the water vapor content. This equilibrium is destroyed if temperature and water vapor content tendencies are perturbed with opposite signs. On the other hand, it cannot be assumed that T and Q have exactly the same error characteristics, as it is supposed in SPPT and pSPPT. Furthermore, in SPPT and pSPPT the wind direction is never altered stochastically, since the tendencies of the U and V components are always using the same stochastic pattern." Page 6, lines 23-25: "The first SPPT version in the IFS model (Buizza et al., 1999) has also used such separate patterns for the different parametrised tendencies. However, it has been removed in the revised SPPT scheme (Palmer et al., 2009) because some physical relationships within a parametrisation scheme could be violated in this way (see Sec. 5)." The ipSPPT is the winner while pSPPT performs very similar to SPPT, except for the surface in January 2017. This latter improvement might be effectively attributed to switching of the tapering function in pSPPT. I would follow the authors that the possibility of altering the wind direction in ipSPPT is a good candidate for explaining the superiority of the ipSPPT in generating reasonable spread, especially in complex terrain like the Alps in winter and for convection in summer. While U and V should be perturbed differently, the T/Q imbalance could be avoided, if these variables are perturbed with the same or correlated stochastic patterns, as it has been described by Christensen et al., 2017 for the handling of processes in their iSPPT scheme: "It is likely that the 'true' errors in the parametrization schemes are neither perfectly correlated as in SPPT nor perfectly uncorrelated as in iSPPT. A further interesting line of

enquiry would be to introduce correlations between the noise patterns used for different parameters. Instead of using two independent patterns in iSPPT, perturbation patterns for the wet processes could be partially correlated with each other, while perturbations for the dry processes could also be partially correlated." Have you tried to modify the SPPT by simply perturbing U, V with different stochastic patterns, or do you know, if anyone else has done this?

————————————————

---

## Short Comment (SC2) · 29 Oct 2018

Dear Editor,

thank you for your comment. The title will be changed in the revised manuscript (including the acronym pSPPT) accordingly to the comment of RC1.

Yours, Clemens Wastl

---

## Referee Comment (RC2) · Anonymous Referee #2 · 9 Nov 2018

Does the paper address relevant scientific modelling questions within the scope of GMD? Yes

Does the paper present a model, advances in modelling science, or a modelling protocol that is suitable for addressing relevant scientific questions within the scope of EGU? Yes

Does the paper present novel concepts, ideas, tools, or data? Yes

Does the paper represent a sufficiently substantial advance in modelling science? Yes

Are the methods and assumptions valid and clearly outlined? Yes

[Figure]

Are the results sufficient to support the interpretations and conclusions? Yes

Is the description sufficiently complete and precise to allow their reproduction by fellow scientists (traceability of results)? Yes

Do the authors give proper credit to related work and clearly indicate their own new/original contribution? Yes

Does the title clearly reflect the contents of the paper? Yes

Does the abstract provide a concise and complete summary? No, the line 13 phrase 'lead to statistically significant improvements...compared to the original SPPT' is not supported by the figures that state statistical significance with respect to an ensemble with no stochastic physics. The abtract or the figures must be corrected in order to be consistent. It is acceptable to mention results that are not statistically significant, but they must be stated as such.

Is the overall presentation well structured and clear? Yes

Is the language fluent and precise? Yes

Are mathematical formulae, symbols, abbreviations, and units correctly defined and used? Yes

Should any parts of the paper (text, formulae, figures, tables) be clarified, reduced, combined, or eliminated? Yes, figures 3-7 must be improved as requested by the other referee.

Are the number and quality of references appropriate? Yes

Is the amount and quality of supplementary material appropriate? Yes

———————————— general comments This is a good paper. It is in the scope of the GMD journal. The research is original and interesting to the ensemble prediction community. It is a significant incremental improvement to the well-known SPPT

scheme. It ties in nicely with related work recently published by Christensen et al.

The presentation is generally clear, scientifically rigorous and well structured.

Specific comments

page 2, line 10: clarify '(e.g. surface fluxes if surface tendencies are not perturbed)' because the SPPT concept could be applied to surface tendencies, although this has not yet been done (for technical reasons).

page 2, line 14: 'destroys the physical consistency' - what do you mean by consistency ? Please clarify or delete. One could simply state that tapering means that model errors near the top and bottom are not represented.

page 2 line 28: 'no interaction of the uncertainties between the schemes is considered'. To be fair, it should be more precisely stated that no interaction _inside the timestep_ is considered. As soon as you perturb the model state, all physics parametrisations will start acting differently at the next timestep. (intuitively the specific originality of iSPPT is that it perturbs physical interactions that occur at timescales shorter than the timestep)

Technical corrections

page 3 line 6: typo 'Meteorlogie' line 10: add a mention to HIRLAM, who are actively developing AROME. line 12: Bénard (with an accent) (I seem to recall Bubnova has one, too) line 24: you mean 'ensemble size', not 'ensemble spread'. Spread and ensemble size are different things, unless you use ensemble size to do something else that produces spread, in which case that 'something else' needs to be stated.

page 4 line 14: append 'at each gridpoint' line 15: correct 'defined by a tapering function (see below)' because you have not defined it yet. line 16: delete the text between parentheses, because it is a repeat of line 7-8 (or the other way around if you prefer). line 18: delete either 'the way' or 'how'.

page 5 line 5: move the sentence about tuning to section 2.3, because it s part of the

experimental setup. line 15: please explain in what sense SPPT was 'unsatisfactory' on the Austrian domain line 19: typo 'diretlcy'

page 7 line 5: typo 'variablese'

page 8 line 11: typo 'adopted' -> 'adapted' or 'adopted in' line 16: rewrite 'supersaturation is translated in a slightly positive temperature increase' (also, 'positive increase' is a tautology)

page 9 line 5: the phrase 'model error is not noticeably influenced' needs to be corrected because (1) model error and forecast error are different things, and (2) error realizations can change even if their statistics do not. You probably mean 'ensemble average error (and/or average member forecast error) is not noticeably changed', since you seem to argue that the reliability of spread is improved by stochastic physics (which would be a valid statement). line 28: insert 'score _differences_ ... in Fig 5 are much smaller'

page 10 line 18: again, replace 'model error' by 'ensemble average error' or 'average forecast error'. line 19-20 replace 'negative CRPS difference' by 'a reduction of CRPS' line 21: replace 'partially' by 'to a lesser extent'

page 11 line 8: replace 'at convective days' by 'on convective days' line 19 insert 'tendencies _of_ U, V, T and Q' line 20 correct 'to _a_ control ensemble' line 24 'reveals a significant increase': a claim of statistical significance is a serious one, so it must be precisely expressed: with respect to what is the increase statistically significant ? SPPT or no SPPT ?

page 12 line 6: insert 'analysis of a _set of_ convective events' line 11: use a more precise term than 'critical', e.g. 'non-consistent', 'physically unsatisfactory', etc. line 12 replace 'switching off tapering' by 'tapering switched off' line 13 grammar '_these_ perturbed fields' line 14: 'could be switched off' is ambiguous', either write 'it is likely that it could be switched off' or 'we have switched it off' line 34: 'is considered critical'

is unclear, do you mean 'unsatisfactory' ?

page 13 line 2: replace 'provides' by 'can provide' because microphysics do not always increase temperature. line 4: you do not really know that 'it cannot be assumed that T and Q have exactly the same error characteristic'. Better write 'it seems wrong to assume that T and Q have exactly the same model error characteristics'.

References: please check the accents for all authors (e.g. Vié etc) a valid web address must be supplied for gray literature references (namely, Palmer et al 2009 and Szucs 2016)

Figures: besides the already mentioned problems with the labelling, the colours for pSPPT and ipSPPT in Fig 3-7 should be made easier to distinguish, perhaps by using a lighter blue for pSPPT. They are impossible to tell apart on two of my screens (not to mention their legibility for most colour-blind readers).

with best regards

—- end of comments —

---

## Author Comment (AC1) · 26 Nov 2018

Dear Referees and Executive editor:

Thank you very much for the review, which is very constructive and helps a lot to improve our manuscript. We have answered your comments, questions etc. point by point in below. Best regards,

Clemens Wastl, and co-authors: 26.11.2018

Please also note the supplement to this comment:

https://www.geosci-model-dev-discuss.net/gmd-2018-184/gmd-2018-184-AC1-supplement.zip

---

## Author Response (AR1)

**Dear Referees:**

Thank you very much for the review, which is very constructive and helps a lot to improve our manuscript. We have answered your comments, questions etc. point by point in below.

Best regards,

Clemens Wastl, and co-authors, 26.11.2018

**Responses to RC1:**

General Comments:

8. Does the title clearly reflect the contents of the paper?
In its present form the title does not really reflect that stochastic perturbations are independently applied to parametrization schemes and prognostic variables.

**The title has been adapted accordingly. "Independent perturbations for physics parametrisation tendencies in a convection permitting ensemble (pSPPT)".**

13. Should any parts of the paper (text, formulae, figures, tables) be clarified, reduced, combined, or eliminated?
Figures 3-7 must be substantially improved. There are no ordinate labels and no references to the different panels in the figure caption.

**Figures 3-7 (including the captions) have been adapted accordingly.**

Specific Comments:

— Abstract: "Both schemes … lead to statistically significant improvements in the probabilistic performance compared to the original SPPT". The authors have shown statistical significance of the different schemes with respect to the reference experiment of no perturbations not with respect to SPPT (see also Page 7, lines 11-12: "The statistical significance of the score differences between the three experiments and the reference run is defined by using a bootstrapping confidence test.")

**This sentence in the abstract has been modified: "… compared to a reference run without stochastic physics."**

— Page 4, line 22: "The tapering function : : : - it is not necessary in some regional models (e.g. WRF, COSMO)." Are there any references? Do you know why?

We do not exactly know the reason for this. The implementations of SPPT in the WRF and COSMO model use especially adapted settings for the perturbations, as stated by Leutbecher et al. (2017): "The differences include the variance of the perturbations, the space and time auto-correlation of the random pattern, the shape of the distribution that is sampled…" We have tried several settings for C-LAEF as well, but without using a tapering function the SPPT implementation was not stable.

**The reference "Leutbecher et al., 2017" has been added to this statement.**

— Page 6, lines 8-10: "A potential drawback of the pSPPT approach is a possible duplication in attributing errors across schemes which can introduce inherent correlations between the perturbations applied to one physics scheme and the output of a later scheme (Christensen et al., 2017)." In contrast to SPPT the pSPPT approach enables switching of the tapering function. Christensen et al. 2017 state about the effects of tapering in SPPT ": : : this method cannot represent uncertainty in the vertical distribution of convective heating. SPPT does not perturb fluxes at the surface or top of atmosphere, introducing inconsistencies between the perturbed tendencies in a column and these fluxes." When balancing the disadvantages from tapering inconsistencies or a possible duplication in attributing errors across schemes, it seems not unreasonable to try out the pSPPT approach. Moreover, it could be expected that errors naturally appear all along the production line of the parametrization schemes and will be processed through the different schemes anyway, presumably being inherently correlated at the output. The main fundamental reason for keeping the parametrization chain deterministic and balanced is that in deterministic parametrisations similar tendencies on input produce similar outputs and thus large scale correlations of the input tendencies on the grid scale will not be altered by the parametrization chain. Therefore, in SPPT only the final output of the chain is perturbed with large scale stochastic patterns which are tied to resolved physical processes on the grid scale. Christensen et al., 2017 state: "SPPT also imposes large spatio-temporal correlation scales when perturbing tendencies to represent the correlation of model uncertainties in space and time, but these correlation scales have not been measured and are not tied to physical processes." Because the correlation length scales of the stochastic patterns are the same one can still expect that the correlations of the input tendencies survive not only the SPPT but also the intermediately perturbed parametrization chain of pSPPT.

Thank you very much for this statement. Yes you are right, we are only using one scale pattern for all the parametrisations and therefore the correlation between input and output tendencies of the complete parametrization chain is kept.

— Page 12, lines 13-15: "Perturbing the physical schemes separately and considering this perturbed fields in the subsequent parametrisation (pSPPT) results in a positive effect on the stability of the model. In this case the tapering function could be switched off for microphysics, radiation, and shallow convection without any problems." Page 12, lines 23-24: "Hence, an interaction of of the uncertainty of one physical scheme in the subsequent one is considered in pSPPT and ipSPPT which seems to increase the consistency of the model," What does consistency of the model means? Does it mean that it runs more stable? It is a little bit surprising that the tapering function could be switched of in the ipSPPT as well, because there is much more uncorrelated noise than in the two other schemes? Where does the stability comes from?

Yes you are right, we are talking about the stability of the model (not consistency). In case of pSPPT and ipSPPT not only the tendencies are passed from one physical parametrisation to the subsequent one, but also the uncertainties (perturbed tendencies). In SPPT the uncertainty is not considered in the physical parametrisations, it is applied at the end of the time step. Hence, in case of pSPPT and ipSPPT the model can react in each parametrisation on the uncertainties coming from the previous one, which is more consistent in our mind than just adding perturbations at the end of the time step.
As written at the end of section 2.2.1 we have used a two weeks test period in 2011 to try different settings of the stochastic schemes and to test the stability of the model when switching off the tapering function. In case of SPPT and no tapering function used we had about 10% model crashes, with ipSPPT about 2% and no crash with pSPPT. This means that ipSPPT is less stable than pSPPT (which is not surprising), but more stable than SPPT, which is of course a bit surprising. Unfortunately we did not look into the details about where the stability in ipSPPT comes from. In our later studies and investigations we focused

completely on the pSPPT approach and did no more consider ipSPPT because as stated in section 2.2.3, some physical relationships within a parametrisation scheme could be violated in this case. But as you say in your last comment, looking only on the scores of this study, it could be worth to enhance this approach.

**We have changed this phrase to: "… seems to increase the stability of the model"**

— Pages 12-13, lines 32-6: "The ipSPPT approach is a modification of pSPPT where the tendencies of the variables T, U, V, and Q receive separate perturbations. As shown in Sect. 3, this approach obtains the best probabilistic scores overall, even though the method is considered critical from a physical point of view. A major concern with the ipSPPT approach is that the balance between the quantities resulting from one parametrisation scheme can be disturbed (Palmer et al., 2009). For example, the microphysics scheme provides an increase of temperature at a certain point due to condensation processes which are also decreasing the water vapor content. This equilibrium is destroyed if temperature and water vapor content tendencies are perturbed with opposite signs. On the other hand, it cannot be assumed that T and Q have exactly the same error characteristics, as it is supposed in SPPT and pSPPT. Furthermore, in SPPT and pSPPT the wind direction is never altered stochastically, since the tendencies of the U and V components are always using the same stochastic pattern." Page 6, lines 23-25: "The first SPPT version in the IFS model (Buizza et al., 1999) has also used such separate patterns for the different parametrised tendencies. However, it has been removed in the revised SPPT scheme (Palmer et al., 2009) because some physical relationships within a parametrisation scheme could be violated in this way (see Sec. 5)." The ipSPPT is the winner while pSPPT performs very similar to SPPT, except for the surface in January 2017. This latter improvement might be effectively attributed to switching of the tapering function in pSPPT. I would follow the authors that the possibility of altering the wind direction in ipSPPT is a good candidate for explaining the superiority of the ipSPPT in generating reasonable spread, especially in complex terrain like the Alps in winter and for convection in summer. While U and V should be perturbed differently, the T/Q imbalance could be avoided, if these variables are perturbed with the same or correlated stochastic patterns, as it has been described by Christensen et al., 2017 for the handling of processes in their iSPPT scheme: "It is likely that the 'true' errors in the parametrization schemes are neither perfectly correlated as in SPPT nor perfectly uncorrelated as in iSPPT. A further interesting line of enquiry would be to introduce correlations between the noise patterns used for different parameters. Instead of using two independent patterns in iSPPT, perturbation patterns for the wet processes could be partially correlated with each other, while perturbations for the dry processes could also be partially correlated." Have you tried to modify the SPPT by simply perturbing U, V with different stochastic patterns, or do you know, if anyone else has done this?

Thank you very much for this interesting statement and the ideas you raised. As mentioned in the paper, from a physical point of view it is not reasonable to perturb T and Q separately because it disturbs the relationship between these two quantities. However, applying different perturbations to the U and V component of wind is very interesting. A colleague from Hungary (Mihaly Szucs) tried this approach for some test cases with the AROME model and the results were quite promising (increase of spread near the surface and no obvious model degradation. The results of this study can be found on
http://www.rclace.eu/File/Predictability/2015/LACE_stay_MihalySzucs2015.pdf
It would be very interesting to compare this approach with the ipSPPT experiment presented in this paper. However, as mentioned in the previous comment, the focus of our later studies was put on the pSPPT approach.

**Responses to RC2:**

General Comments:

Does the abstract provide a concise and complete summary? No, the line 13 phrase 'lead to statistically significant improvements...compared to the original SPPT' is not supported by the figures that state statistical significance with respect to an ensemble with no stochastic physics. The abstract or the figures must be corrected in order to be consistent. It is acceptable to mention results that are not statistically significant, but they must be stated as such.

**The sentence in the abstract has been modified accordingly. "… compared to a reference run without stochastic physics."**

Should any parts of the paper (text, formulae, figures, tables) be clarified, reduced, combined, or eliminated? Yes, figures 3-7 must be improved as requested by the other referee.

**Figures 3-7 (including the captions) have been adapted accordingly.**

Specific Comments:

page 2, line 10: clarify '(e.g. surface fluxes if surface tendencies are not perturbed)' because the SPPT concept could be applied to surface tendencies, although this has not yet been done (for technical reasons).

**The phrase "… if surface tendencies are not perturbed …" has been added to this sentence.**

page 2, line 14: 'destroys the physical consistency' - what do you mean by consistency? Please clarify or delete. One could simply state that tapering means that model errors near the top and bottom are not represented.

**We replaced physical consistency with "… physical consistent representation of model uncertainty in the vertical …"**

page 2 line 28: 'no interaction of the uncertainties between the schemes is considered'. To be fair, it should be more precisely stated that no interaction _inside the timestep_ is considered. As soon as you perturb the model state, all physics parametrisations will start acting differently at the next timestep. (intuitively the specific originality of iSPPT is that it perturbs physical interactions that occur at timescales shorter than the timestep)

Thank you very much for this comment. You are right, pSPPT and ipSPPT enable an interaction of uncertainties between the different parametrisations within one timestep, whereas in SPPT the model acts on the perturbations in the next timestep.

**The phrase "… inside a timestep …" has been added.**

Technical corrections:

page 3 line 6: typo 'Meteorlogie'

**Corrected.**

page 3 line 10: add a mention to HIRLAM, who are actively developing AROME.

**We added the phrase "… HIRLAM (High Resolution Limited Area Model, Bengtsson et al., 2017) …" and the corresponding publication to the reference list.**

page 3 line 12: Bénard (with an accent) (I seem to recall Bubnova has one, too)

**Changed, also in the reference list.**

page 3 line 24: you mean 'ensemble size', not 'ensemble spread'. Spread and ensemble size are different things, unless you use ensemble size to do something else that produces spread, in which case that 'something else' needs to be stated.

Yes you are right, we are meaning ensemble size.

**Phrase changed to "… good compromise between ensemble size and computational costs."**

page 4 line 14: append 'at each gridpoint'

**Done.**

line 15: correct 'defined by a tapering function(see below)' because you have not defined it yet.

**Done.**

line 16: delete the text between parentheses, because it is a repeat of line 7-8 (or the other way around if you prefer).

**Done.**

line 18: delete either 'the way' or 'how'.

**Done.**

page 5 line 5: move the sentence about tuning to section 2.3, because it s part of the experimental setup.

**Done.**

page 5 line 15: please explain in what sense SPPT was 'unsatisfactory' on the Austrian domain

**The phrase "… which have produced unsatisfactory results within the Austrian domain …" has been removed.**

page 5 line 19: typo 'diretlcy'

**Corrected.**

page 7 line 5: typo 'variablese'

**Corrected.**

page 8 line 11: typo 'adopted' -> 'adapted' or 'adopted in'

**Corrected.**

page 8 line 16: rewrite 'supersaturation is translated in a slightly positive temperature increase' (also, 'positive increase' is a tautology)

**The word "positive" has been removed.**

page 9 line 5: the phrase 'model error is not noticeably influenced' needs to be corrected because (1) model error and forecast error are different things, and (2) error realizations can change even if their statistics do not. You probably mean 'ensemble average error (and/or average member forecast error) is not noticeably changed', since you seem to argue that the reliability of spread is improved by stochastic physics (which would be a valid statement).

Yes you are right, we are talking about ensemble average error.

**Changed to "… ensemble average error …".**

page 9 line 28: insert 'score _differences_ ... in Fig 5 are much smaller'

**Done.**

page 10 line 18: again, replace 'model error' by 'ensemble average error' or 'average forecast error'.

**Done.**

page 10 line 19-20 replace 'negative CRPS difference' by 'a reduction of CRPS'

**Done.**

page 10 line 21: replace 'partially' by 'to a lesser extent'

**Done.**

page 11 line 8: replace 'at convective days' by 'on convective days'
**Done.**

page 11 line 19 insert 'tendencies _of_ U, V, T and Q'

**Done.**

page 11 line 20 correct 'to _a_ control ensemble'

**Done.**

page 11 line 24 'reveals a significant increase': a claim of statistical significance is a serious one, so it must be precisely expressed: with respect to what is the increase statistically significant ? SPPT or no SPPT ?

Thank you for this comment. The ensemble spread is statistically significantly increased compared to the reference experiment (Fig.6d).

**We added "… statistically significant increased ensemble spread compared to the reference experiment."**

page 12 line 6: insert 'analysis of a _set of_ convective events'

**Done.**

page 12 line 11: use a more precise term than 'critical', e.g. 'non-consistent', 'physically unsatisfactory', etc.

**Thank you. We changed "critical" to "physically unsatisfactory".**

page 12 line 12 replace 'switching off tapering' by 'tapering switched off'

**Done.**

page 12 line 13 grammar '_these_ perturbed fields'

**Corrected.**

page 12 line 14: 'could be switched off' is ambiguous', either write 'it is likely that it could be switched off' or 'we have switched it off'

**The sentence has been changed to "…the tapering function has been switched off…".**

page 12 line 34: 'is considered critical' is unclear, do you mean 'unsatisfactory' ?

Yes we are meaning unsatisfactory.

**We changed "critical" to "unsatisfactory".**

page 13 line 2: replace 'provides' by 'can provide' because microphysics do not always increase temperature.

**Changed.**

page 13 line 4: you do not really know that 'it cannot be assumed that T and Q have exactly the same error characteristic'. Better write 'it seems wrong to assume that T and Q have exactly the same model error characteristics'.

**Changed.**

References: please check the accents for all authors (e.g. Vié etc) a valid web address must be supplied for gray literature references (namely, Palmer et al 2009 and Szucs 2016).

**Done.**

Figures: besides the already mentioned problems with the labelling, the colours for pSPPT and ipSPPT in Fig 3-7 should be made easier to distinguish, perhaps by using a lighter blue for pSPPT. They are impossible to tell apart on two of my screens (not to

mention their legibility for most colour-blind readers).

**Thank you for this comment. We changed the color for pSPPT to a lighter blue.**

**Responses to SC1:**

General Comments:

"The main paper must give the model name and version number (or other unique identifier) in the title."
Please add at least the acronym of your new parametrisation (pSPPT) in the title of your manuscript upon revision of the article. Additional, if applicable, add a version number (the method might be changed in the future and those methods should be distinguishable).

**The title has been adapted accordingly. "Independent perturbations for physics parametrisation tendencies in a convection permitting ensemble (pSPPT)".**

In addition, regarding the code availability section, in case it is possible for the user to obtain a license, please add further information on how to obtain it. Be aware, that in all cases the exact version of your method published here, must be kept.

**We have added the email address to obtain a license agreement with ZAMG to the code availability section.**

[revised manuscript text omitted]

15    Soc., 99, 1415-1432, doi:10.1175/BAMS-D-16-0321.1, in press, 2018.

Weidle, F., Wang, Y., Tian, W., Wang, T.: Validation of strategies using Clustering analysis of ECMWF-EPS for initial perturbations in a Limited Area Model Ensemble Prediction System. Atmosphere-Ocean, 51, 248-295, doi: http://dx.doi.org/10.1080/07055900.2013.802217, 2013.

Weisheimer, A., Corti, S., Palmer, T. N., and Vitart, F.: Addressing model error through atmospheric stochastic physical

20    parametrizations: impact on the coupled ECMWF seasonal forecasting system. Philos. Trans. R. Soc. A, 372, doi: 10.1098/rsta.2013.0284, 2014.

Wilks, D.: Statistical Methods in the Atmospheric Sciences, Volume 100 3rd Edition, Academic Press, 704 pp, 2011.

[Figure]

**Figure 1. Domain of the C-LAEF system including the INCA domain for precipitation verification (red). The colouring shows the altitude (m).**

[Figure]

**Figure 2. Illustration of how the stochastic perturbations are applied in the different physics parametrization schemes of SPPT (first row), pSPPT (second row) and ipSPPT (last row).**

[Figure]

**Figure 3. Ensemble spread (solid lines) and RMSE (dashed lines) as a function of lead time for temperature at 500 hPa (a), 850 hPa (b, e) and wind speed at 500 hPa (c), 00 hPa  (d, f) in July 2016. Panels a to d  are shown as difference to an ensemble without any stochastic physics (REF), circles (crosses) denote significant differences for the ensemble spread (RMSE). Panels e and f show absolute numbers for all four experiments at 850 hPa.**

[Figure]

**Figure 4.**  Continuous Ranked Probability Score (CRPS) as a function of lead time for surface variables 2 m temperature (a), 10 m wind speed (b), mean sea level pressure (c) and precipitation () in July 2016.  Panels e and f shows the BIAS of 2 m temperature  and relative humidity  for the same period. All numbers are shown as a difference to C-LAEF without any stochastic physics (REF). Circles denote significant differences of CRPS and BIAS, respectively.

[Figure]

Figure 5. Ensemble spread (solid lines) and RMSE (dashed lines) as a function of lead time for temperature at 500 hPa (Ka), 850 hPa (b) and wind speed at 500 hPa (m s-1c), at 500 hPa and 850 hPa (d) in January 2017. Scores are shown as difference to an ensemble without any stochastic physics (REF), circles (crosses) denote significant differences for the ensemble spread (RMSE).

[Figure]

**Figure 6. As in  Fig. 3, but for surface variables 2 m temperature (a), 10 m wind speed (b), mean sea level pressure (c) and precipitation (d) in January 2017. The last row shows absolute numbers for temperature (e) and precipitation (f).**

[Figure]

**Figure 7. Ensemble spread (a, b) and RMSE (c, d) of precipitation  as a function of lead time. The panels a and c refer to days with high convective activity, the panels b and d to days with stable conditions. Scores are shown as difference to an ensemble without any stochastic physics (REF).**